# A Deep Learning Framework for the Prediction and Diagnosis of Ovarian Cancer in Pre- and Post-Menopausal Women

**DOI:** 10.3390/diagnostics13101703

**Published:** 2023-05-11

**Authors:** Blessed Ziyambe, Abid Yahya, Tawanda Mushiri, Muhammad Usman Tariq, Qaisar Abbas, Muhammad Babar, Mubarak Albathan, Muhammad Asim, Ayyaz Hussain, Sohail Jabbar

**Affiliations:** 1Department of Electrical Engineering, Harare Polytechnic College, Causeway Harare P.O. Box CY407, Zimbabwe; 2Department of Electrical, Computer and Telecommunications Engineering, Botswana International University of Science and Technology, Palapye 10071, Botswana; yahyaa@biust.ac.bw; 3Department of Industrial and Mechatronics Engineering, Faculty of Engineering & the Built Environment, University of Zimbabwe, Mt. Pleasant, 630 Churchill Avenue, Harare, Zimbabwe; tmushiri@medsch.uz.ac.zw; 4Abu Dhabi University, Abu Dhabi 59911, United Arab Emirates; 5College of Computer and Information Sciences, Imam Mohammad Ibn Saud Islamic University (IMSIU), Riyadh 11432, Saudi Arabia; qaabbas@imamu.edu.sa (Q.A.); mmalbathan@imamu.edu.sa (M.A.);; 6Robotics and Internet of Things Laboratory, Prince Sultan University, Riyadh 12435, Saudi Arabia; mbabar@psu.edu.sa; 7EIAS Data Science Laboratory, Prince Sultan University, Riyadh 12435, Saudi Arabia; masim@psu.edu.sa; 8Department of Computer Science, Quaid-i-Azam University, Islamabad 44000, Pakistan; ayyaz.hussain@qau.edu.pk

**Keywords:** prediction, diagnosis, epithelial ovarian cancer, histopathological images, convolutional neural networks, augmentation

## Abstract

Ovarian cancer ranks as the fifth leading cause of cancer-related mortality in women. Late-stage diagnosis (stages III and IV) is a major challenge due to the often vague and inconsistent initial symptoms. Current diagnostic methods, such as biomarkers, biopsy, and imaging tests, face limitations, including subjectivity, inter-observer variability, and extended testing times. This study proposes a novel convolutional neural network (CNN) algorithm for predicting and diagnosing ovarian cancer, addressing these limitations. In this paper, CNN was trained on a histopathological image dataset, divided into training and validation subsets and augmented before training. The model achieved a remarkable accuracy of 94%, with 95.12% of cancerous cases correctly identified and 93.02% of healthy cells accurately classified. The significance of this study lies in overcoming the challenges associated with the human expert examination, such as higher misclassification rates, inter-observer variability, and extended analysis times. This study presents a more accurate, efficient, and reliable approach to predicting and diagnosing ovarian cancer. Future research should explore recent advances in this field to enhance the effectiveness of the proposed method further.

## 1. Introduction

The yearly mortality for ovarian cancer is 151,900, making it the deadliest cancer globally [1]. According to Miller, it is women’s fifth highest cause of death. The most common kind of gynaecological carcinoma is ovarian cancer, which emanates from epithelial tissue, and 90% of the cases are due to this type. The five histologic carcinomas are Mucinous-Ovarian Cancer (M-OC), High-Grade Serous-Ovarian Cancer (H-GS-OC), Low-Grade Serous-Ovarian Cancer (L-GS-OC), Clear-Cell Ovarian Cancer (C-COC), and Endometrioid-Ovarian-Cancer (E-O-C), with poor prognosis at an advanced stage [2].

Early Identification boosts survival from 3% in Stage IV to about 90% in Stage I [3]. Van Haaften-Day et al. (2001) [4] found that Carbohydrate Antigen 125 (CA125) has been used for over four decades, but that its accuracy is not acceptable as it has not improved survivability. 50% of early-stage tumours, primarily type I ovarian cancers, and 92% of advanced-stage tumours, primarily type II ovarian cancers, have increased serum CA125 levels. Skates et al. [5] found that physiological variables influence normal CA125 serum concentrations and the presence of menopause impacts CA125 levels. Sopik et al. [6] found that benign illnesses also had increased CA125 levels, causing false positives. Only 20% of ovarian tumors express CA125. A screening’s anticipated sensitivity is around 80%. Moss et al. [7] confirmed that relying on the biomarker alone is misleading. Akinwunmi et al. [8] found that 1% of healthy people had increased serum levels, including 5% of patients with benign illnesses, e.g., endometriosis.

HE4, WFDC2, is overexpressed in Endometrioid ovarian cancer and less in epithelial tissues of the respiratory system and reproductive organs [9]. Yana-ranop et al. [10] found that the specificity of HE4 was 86% compared to that of CA125, and the AUC of it was 0.893 compared to that of CA125 0.865 [11]. HE4 levels vary in smokers (30%) and non-smokers (20%) [12]. Contraceptives impact HE4 levels. Ferraro et al. [13] found lower HE4 levels in oral contraceptive users (*p* = 0.008). Biopsy, imaging (US, CT, MRI, PET), and algorithms for learning (deep) Convolutional Neural networks (CNN) can predict and diagnose epithelial ovarian cancer (serous) accurately.

The challenge is that there is no effective screening method; hence, ovarian cancer is diagnosed when it has already advanced to Stage III or IV. Radiologists manually analyse and interpret medical images of a suspected cancer patient for cancer subtyping and staging. This results in the misclassification of the cancer subtypes, inter-observer variations, subjectivity, and time consumption. To address this, a deep CNN model was developed to predict and diagnose cancers.

Expert pathologists interpret Cellular morphology, defining OC categories and directing treatments [14]. Inter-observer variations cause inaccurate, suboptimal treatments, poor prognosis, and reduced life quality [15]. This shows the need for computational methods to accurately predict cancer class and diagnose cancer subtype.

Accurate prediction and diagnosis of the cell tumours are vital as they lead to proper prognosis and treatment, increasing survivability. Deep learning merits include:Processing huge data and producing highly accurate predictions, reducing incorrect diagnoses.Permitting early detection of ovarian carcinoma, increasing treatment success.Permitting personalised treatment. Deep learning can predict how treatments affect different women, enabling personalised, efficient care.

Deep learning can improve patient outcomes and reduce mortality through early detection and personalised treatment. The algorithm could predict and diagnose the images in under 5 s with an F1-score of 0.94. The order of the paper is as follows: related work, the materials and methods used, followed by results and discussion, and, finally, the conclusion.

## 2. Related Work

Various deep-learning techniques have been studied to classify ovarian carcinoma based on the cell type. Personalized treatment plans for ovarian cancer patients depend on accurately identifying the type of ovarian cancer. In the past decade, various studies have aimed to enhance cancer screening outcomes in the preclinical stage, focusing on utilizing histopathological images and biomarkers, such as CA-125 and HE-4. However, biomarker-based detection can be slow and prone to missing detection [16,17]. CA-125 has limited accuracy and specificity in detecting early-stage ovarian cancer, and imaging methods such as CT, US, and MRI are commonly used for locating and detecting features of masses. However, the interpretation of images by expert radiographers can be time-consuming and prone to inter-observer variability [18].

Recent studies have focused on using machine learning algorithms for early prediction and diagnosis of ovarian cancer. Different methods have been proposed to extract features from ultrasound images and classify ovarian tumors. Some techniques include SVM, shallow neural network classifiers, and wavelet transform filters. The features extracted from the images include textural and pathological features, wavelet coefficients, homogeneousness, histogram, and grey difference multi-scaling. These features are then integrated with a support-vector-machine (SVM) classifier to classify the types of cysts fully [19,20,21,22,23,24]. El-Bendary and Belal [25] developed a combined approach for subtyping cancer stage and ovarian cancer classification using gene expression and clinical datasets. Boosting and Ensemble SVM achieved a classification accuracy of 80%, while other ML classifiers yielded 70.77%. The proposed method also had a high recall, specificity, precision, F measure, and AUC.

The authors in [26] developed a CAD method for diagnosing borderline ovarian cancer by analyzing S-HG images. They used a k-NN classifier, and an optimization tool called T-POT, which resulted in an average accuracy between 0.976 and 0.96. The T-POT classifier achieved AUROC values of 0.97, 1, 0.98, and 0.99 for normal, malignant, borderline, and benign cases, respectively.

Deep learning techniques offer the advantage of intricate feature learning directly from raw data, making it possible to define a system without manually creating features that are necessary for other machine learning methods. This property has drawn interest in examining the benefits of using deep learning in medical image processing [27,28]. Multiphoton Microscopy (MPM) imaging and wide-field fluorescence imaging (WFI) has been proposed for ovarian imaging to capture specific biomarkers that cannot be detected by MPM alone while providing high-resolution images of the entire ovary. Using the Linear Discriminant Analysis classification algorithm, Sawyer et al. [29] achieved an accuracy of 66.66%, 87.50%, and 62.5% for genotype, age, and treatment, respectively [30,31].

The article highlights the different methods researchers propose for detecting and classifying various types of cancer using deep learning algorithms. Lu et al. [32] proposed a Tumour Origin and Assessment method via Deep Learning (TOAD), yielding an AUC-ROC of 0.988. Booma et al. [33] introduced a method using max-pooling enhanced with ML algorithms, which achieved an accuracy of 89%. Wen et al. utilized a custom collection of 3D filters achieving accuracy with AU-ROC ranging from 83% to 90% [34] (Wen et al., 2016). Huttunen et al. [35] used deep neural networks to classify unstained tissue multiphoton microscopy images achieving 95% sensitivity and 97% specificity. Wang et al. [36] proposed a two-level deep transfer learning approach, achieving an accuracy of 87.54%. Lastly, Yu et al. [37] proposed a radiomics model based on a convolutional neural network (CNN), which achieved an AUC of 0.955 to 0.975. They suggested that a CNN-based radiomics method for predicting endometrial cancer and distinguishing histopathology slides with cancer cells, showing potential clinical value in detecting tumour cells and predicting chemotherapy response.

Sengupta et al. [38] used DHL with lamin protein distribution to diagnose ovarian cancer based on the morphology of the nucleus, showing potential as a useful marker in prognosis and histology. Zhang et al. [39] used radiomics and a CNN-based model to predict Endometrioid Cancer with promising clinical potential, although with a limited sample size. Liao et al. [40] proposed the MTDL method for addressing high-dimensional feature space issues and improving classification accuracy, with the potential for further application to additional dataset categories. Various studies have proposed promising methods for diagnosing and predicting gynecological cancers, including ovarian and endometrial cancers, using advanced techniques such as MPM, WFI, DHL, radiomics, and CNN-based models, which hold potential for improving clinical outcomes.

Guo et al. [41] used deep and machine-learning techniques for clustering and classification in ovarian cancer subtyping using gene expression features. Kasture et al. [42] proposed a DL approach for predicting and classifying ovarian cancer subtypes using histopathological images. The authors in [43] used F-R-CNN to classify ovarian images. Ghoniem et al. [44] proposed a hybrid DL using multimodal data, combining gene and histopathology images with ALO-optimized LSTM and CNN networks. The proposed study of Ghoniem et al. suggests a novel approach to diagnosing ovarian cancer using a hybrid evolutionary deep-learning model that integrates multiple types of data and various assessment indicators. By comparing the performance of the proposed model with nine other multi-modal fusion models, the study demonstrates that the proposed model achieves higher precision and accuracy in ovarian cancer diagnoses.

A visual example of early detection through conventional methods as shown in Figure 1.

Finally, Xao et al. [45] used multi-omics to identify biomarkers for early OC diagnosis. We requested readers to check out in details the figure [46], which shows women with genetic mutations, or a family cancer history are at high risk. Tissue and liquid biopsy were analyzed using omics technology, combined with ML technology for biomarker development, and combined with the machine learning algorithm to form omics [47]. In addition, the authors refer to figure [47], which relates to using multi-omics to identify biomarkers, which speeds up the process of discovering new biomarkers. The process is displayed in Figure 2.

In their seminal work, Arezzo et al. investigated the application of radiomics in the qualitative and quantitative analysis of images obtained from various imaging techniques, such as magnetic resonance imaging (MRI) and ultrasound [48]. The primary objective of the radiomics approach is to achieve personalized ovarian cancer medicine. However, they concluded that radiomics has not yet been widely adopted in clinical settings due to several challenges, including:
The absence of a benchmark protocol for feature extraction in each method involved.Bias is introduced by differences in images acquired from various instruments.A lack of prospective external validation of algorithmic models on large datasets, resulting in insufficient studies addressing the performance of radiomics [48]. Arezzo et al. emphasized that images cannot only show the extent of the disease but also aid in its diagnosis. They suggested that deep learning could potentially overcome some of the limitations of radiomics in image analysis.

In a subsequent study, Arezzo et al. [48] revealed that peritoneal tuberculosis (TBP) exhibits similar signs and symptoms to advanced ovarian cancer. TBP accounts for approximately 1% to 2% of tuberculosis cases. The similarity between TBP and ovarian carcinoma presents challenges in the differential diagnosis. Biopsy remains the definitive method for confirming the presence of the disease, while computed tomography (CT) and ultrasound can be employed for visualizing nodules [49]. The authors proposed that a deep learning framework could effectively enhance the accuracy of image analysis, providing more objective diagnoses when using these imaging techniques.

Reilly et al. [50] outline the development and validation process of MIA3G, a deep neural network algorithm that aims to detect ovarian cancer. The algorithm was trained on a dataset of 1067 specimens and validated on a separate set of 2000 specimens. The results show that MIA3G achieves a sensitivity of 89.8% and a specificity of 84.02% in detecting ovarian cancer.

Machine learning integrates and analyses high-throughput molecular experiments to create new biomarkers for understanding illness. Integrated data analysis through machine learning can reveal biologically meaningful biomarker candidates despite changes at each omics level. Artificial intelligence studies are predicted to improve precision medicine for OC significantly. Elyan et al. [51] reviewed advances in using deep learning to analyse images and compared ML and DL algorithms. Figure 3 compares machine learning techniques that require manual feature extraction and deep learning, which do the extraction automatically.

Machine learning (ML) [53] enables computers to learn from past data without creating algorithms to account for the unlimited number of possible feature combinations. DL techniques differ from traditional approaches that use manual labelling. DL techniques learn the features of incoming photos from start to end without feature extraction or engineering. Convolutional neural networks (C-NNs) can capture the underlying representation of images by using partially linked layers and weight sharing, as shown in Figure 4.

The studies highlighted the potential of deep learning and machine learning techniques in ovarian cancer subtyping and diagnosis using various data modalities, including gene expression, histopathological images, and ultrasound images. These methods have shown promising results in improving accuracy and identifying subtypes at the molecular level. However, the small sample sizes and the need for larger datasets remain a knowledge gap.

Table 1 compares machine learning methods used for diagnosing and subtyping ovarian cancer. The table lists various researchers’ studies, methodologies, and performance metrics, such as accuracy, sensitivity, specificity, and AUC-ROC.

## 3. Materials and Methods

This section discusses the methodology used in the research, including the dataset preparation and the proposed Convolutional Neural Network (CNN) architecture. The dataset used in this project consists of 200 images, with 100 images each of serous ovarian cancer and non-cancerous samples. The original dataset was obtained from The Cancer Genome Atlas TCGA repository and was augmented to 11,040 images for both classes to be effectively used in the deep learning architecture (Kasture et al., 2021) [42]. This dataset was split into 80% for the training set and 20% for validation. The proposed CNN architecture is shown in Figure 5.

The preprocessed dataset was then fed for convolution operation. The preprocessing conducted on the images was the elimination of the images that had errors, such as the ones which were encoded badly and did not have the jpeg extension. The data was cleaned to remove the images that were distorted. Convolution is a special line function used to extract a feature using some feature detectors, also called a kernel, which are applied to all inputs, and a series of numbers called a tensor. The product-wise element between each kernel element and the input tensor is calculated in each tensor area and summarized to find the output value in the corresponding outgoing tensor area called the feature map. With striding and padding included, the formula for determining the output image dimension was as follows:(1)$$\mathrm{output}\mathrm{image}\mathrm{dimension}=\left[\frac{n+2p-f}{s}+1\right]\;\times \;\left[\frac{n+2p-f}{s}+1\right]$$
where

*n* = Number of input pixels*f* = Number of pixels of filter*p* = Padding*s* = Stride

To mimic the real-world representation mathematically of the behavior of neurons, the result of convolution was passed on to the non-linear activation function, ReLU, defined mathematically as $f\left(x\right)=max(0,x)$ The activation function choice includes the fact that it does not engage all of the neurons at the same time and does not stimulate all of the neurons at the same time. Hence during backpropagation, not all the neurons are activated. Next, the data was passed onto the pooling layer, which provided a standard downtime that reduced the size of the feature map element to introduce translation flexibility in tiny shifts and distortions, and that reduced the number of subsequent parameters that could be learned. The research used Maxpooling. The data was then fed into the flattening layer and converted into a one-dimensional array or (vector) numbers for inputting into the fully connected layer. After that, the data is fed to the fully connected layer of feedforward neural networks. Every node in the previous layer is linked to the next layer’s nodes by a learnable weight. The features from the pooling layers are mapped to the network’s outputs. Lastly, the result was fed to the output layer, which used SoftMax for classification. The Loss function used was cross entropy and was given by the following formula for binary classification:(2)$$l=-\left(ylog\left(p\right)+\left(1-y\right)log\left(1-p\right)\right)$$
where:*l* = Loss function*p* = The predicted probability*y* = 0 or 1 in binary classification

In conclusion, the proposed CNN architecture and dataset preparation for ovarian cancer classification has been discussed. The algorithm used to train the augmented image dataset has been summarized, and Table 2 summarizes the hyperparameters used in the CNN architecture.

## 4. Results

The model was fed with 11,040 images––half healthy cells, and the other half infected with serous cancer subtype. The number of epochs was chosen for hyperparameter selection and incremented in tens. The training and validation accuracy were recorded.

Table 3 shows a steady increase in the training and validation accuracy as the number of epochs increased from 10 to 50, reaching a training accuracy of 99.52% and a validation accuracy of 99.91% after 50 epochs.

After the training process, the testing process was performed by uploading an image from the test dataset, and the algorithm would give the percentage of it being either serous or healthy cells. This result demonstrates the superiority of the Xception network, which achieves high training and validation accuracy with a small number of epochs. Compared to conventional generic convolutional neural networks, the model does not perform channel-wise convolution, reducing the number of connections and making the model lighter. As a result, excellent accuracy can be achieved with just 10 epochs (99.03% training accuracy and 94.43% validation accuracy).

The model accuracy shows that the percentage for training and validation accuracy rose as shown in Figure 6. This graph shows accuracy from 98.56% and 99.73% to 99.52% and 99.91% for 10 to 50 epochs, respectively.

The percentage loss (shown in Figure 7) for training and validation stood at 0.0445 and 0.0083 and 0.0147, and 0.0020 after 10 and 50 epochs, respectively.

Figure 8 shows the 93.02% of the images were correctly classified as health, and 95.12% were classified as serous cells. From the confusion matrix, valuable performance metrics can be derived, such as:
(3)$$\mathrm{Sensitivity}=\frac{TP}{TP+FN}=\frac{93.02}{93.02+4.88}=0.9502$$

The sensitivity of the model is 0.9502, which represents the model’s ability to predict and classify the health cell images correctly as health when the label class is health.
(4)$$\mathrm{Specificity}=\frac{TN}{TN+FP}=\frac{95.12}{95.12+6.98}=0.9316$$

The specificity represents the model’s ability to correctly predict and classify the infected cells as infected cells when the class label is serous.
(5)$$\mathrm{Recall}=\frac{TP}{TP+FN}=0.9502$$

Recall measures how well the model can detect positive events, as in the health images class detection.
(6)$$\mathrm{Precision}=\frac{TP}{TP+FP}=\frac{93.02}{93.02+6.98}=0.9302$$

Precision measures how well the model has assigned the positive events to the positive class.
(7)$$\mathrm{The}\;F\text{-}\mathrm{measure}=2\times \frac{Recall\times Precision}{Recall+Precision}=2\times \frac{0.9502\times 0.9302}{0.9502+0.9302}=0.94$$

The F-score is the harmonic mean of the model’s precision and recall, and it measures the model’s accuracy on the dataset of health and serous.

Table 4 shows that the precision for health is higher than that of serous cells by 2 units, and the recall for both is equal at 0.93. The F1 score is 0.94 for all classes. The classification report summarizes the performance of the Xception network on the medical images for the entire dataset.

Table 5 shows the accuracy score for different classification models; deep hybrid learning, convolutional neural network, GoogleNet (v3), and linear discriminant analysis classification. The GoogleNet v3 achieved the highest accuracy of 0.925, and the linear discriminant achieved the lowest of 0.666. As a comparison, the accuracy of GoogleNet v3 can be attributed to the transfer learning process, whereby the features are input to a pre-trained architecture. The hyperparameters would have been tuned already. The method that was proposed outperformed the ones by other researchers. A classification accuracy of 0.94 can be attributed to the efficient use of the parameters. The network is built because it has batch normalization after every convolution, which helps fight the model’s overfitting.

Table 5 shows that a “Deep Hybrid Learning” model achieved an accuracy score of 0.907. This model combines deep learning and traditional machine learning techniques, such as decision trees or linear regression, to improve performance. The specific structure of the network used in the study is not provided. Still, it likely involves multiple layers of deep neural networks and incorporates techniques such as batch normalization to combat overfitting.

## 5. Discussion

In this study, we aimed to develop a deep-learning model to classify healthy and serous ovarian cancer cells accurately. Our model achieved a high accuracy of 94.43%, a sensitivity of 0.9502, and a specificity of 0.9316, demonstrating its potential to assist physicians in diagnosing and predicting ovarian cancer prognosis, ultimately improving patient outcomes.

Our findings build upon previous research in the field, with our model surpassing the performance of other models reported in the literature. The superior performance can be attributed to the depth-wise separable convolution, which reduces model complexity, and the absence of activation functions in intermediate layers and residual connections. These innovations enabled our model to achieve higher accuracy than the 80% reported by El-Nabawy et al. [25], the 66.66–87.5% by Sawyer et al. [29], and the 83.6% by Lu et al. [32].

However, the study has some limitations. A public TCGA dataset may not fully represent local patient populations, and the retrospective design does not account for disease progression. Additionally, the lack of clinical data, such as patient genetics, may affect model robustness. To address these issues, future research should involve partnerships with hospitals and research centers to validate the model on real-world data and incorporate clinical datasets.

Alternative explanations for the findings should also be considered. Model interpretation techniques could provide insights into the underlying factors driving the model’s predictions, and the model’s design can be further refined to improve performance.

The practical implications of our findings include the potential to support ovarian cancer diagnosis and monitoring in clinical settings. Our model also offers recommendations for future research, such as addressing limitations, improving model interpretability, and validating the model on diverse populations.

## 6. Conclusions

In summary, this study explored the potential of a deep learning model for predicting and diagnosing ovarian cancer. Our model achieved an accuracy of 94%, highlighting its promise for early prediction and diagnosis. The study contributes to the existing literature by demonstrating the potential of deep learning in preclinical cancer screening and personalized management of ovarian cancer.

The broader implications of our research include the potential impact on related fields, such as radiology, and real-world applications in ovarian cancer diagnosis and treatment optimization. Despite some limitations, our study offers suggestions for future research to build upon our findings, such as improving data quality and quantity, evaluating models on new independent datasets, and enhancing generalizability through diverse population studies.

Overall, this research emphasizes the potential of advanced computing techniques in the timely detection of ovarian cancer. By addressing current challenges and refining these models, we can develop more reliable and broadly applicable tools that can be adopted for practical medical purposes, ultimately improving cancer diagnosis, outcome prediction, and personalized medical care.

## Figures and Tables

**Figure 1 diagnostics-13-01703-f001:**
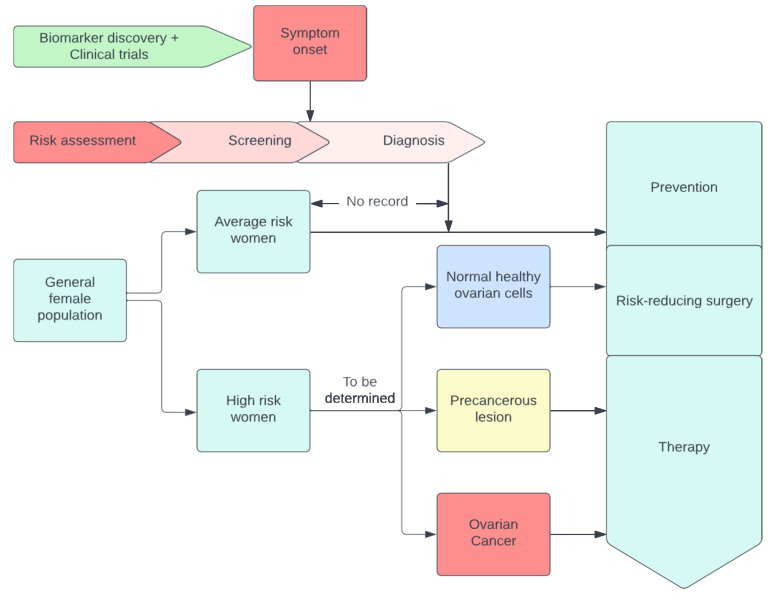
A visual example of early detection through conventional methods.

**Figure 2 diagnostics-13-01703-f002:**
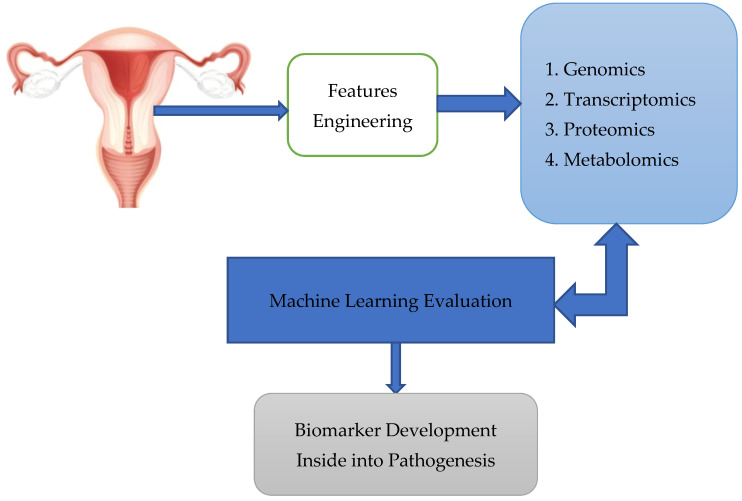
Tissue and liquid biopsy were analysed using omics technology, combined with ML technology for biomarker development, combined with the machine learning algorithm to form omics.

**Figure 3 diagnostics-13-01703-f003:**
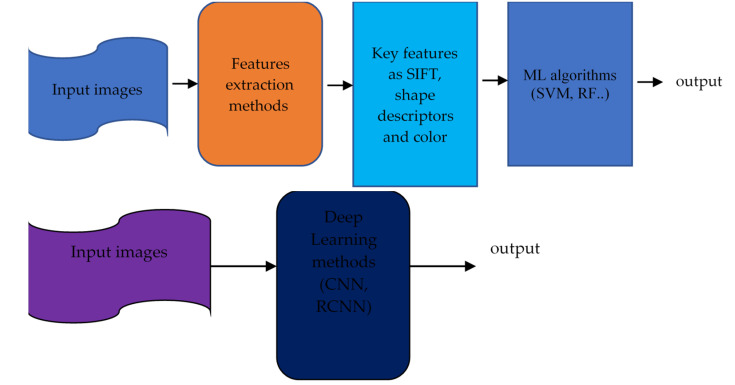
The approach of Earlier C-V techniques (**top**) against DL (**bottom**). (Gumbs et al.) [52].

**Figure 4 diagnostics-13-01703-f004:**
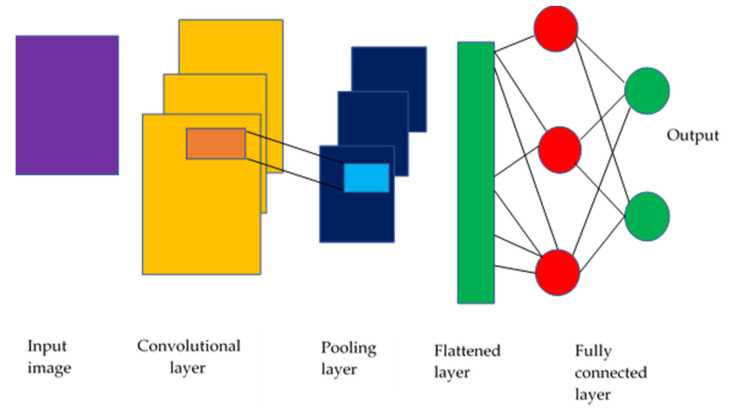
CNN model schematic (Sarvamangala & Kulkarni, 2022) [53].

**Figure 5 diagnostics-13-01703-f005:**
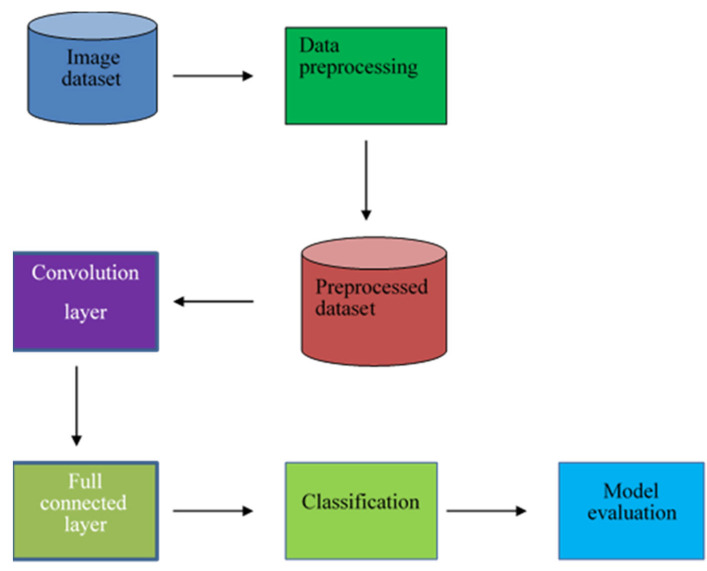
The proposed architecture of the model.

**Figure 6 diagnostics-13-01703-f006:**
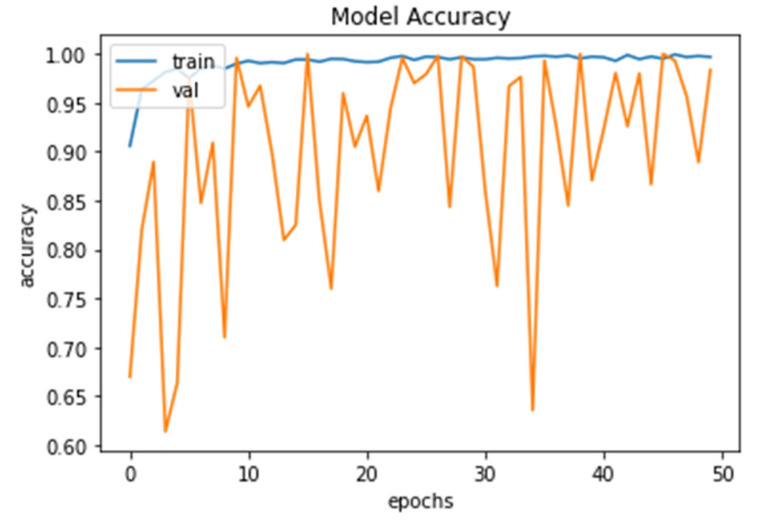
Model Accuracy for both training and validation after 50 epochs.

**Figure 7 diagnostics-13-01703-f007:**
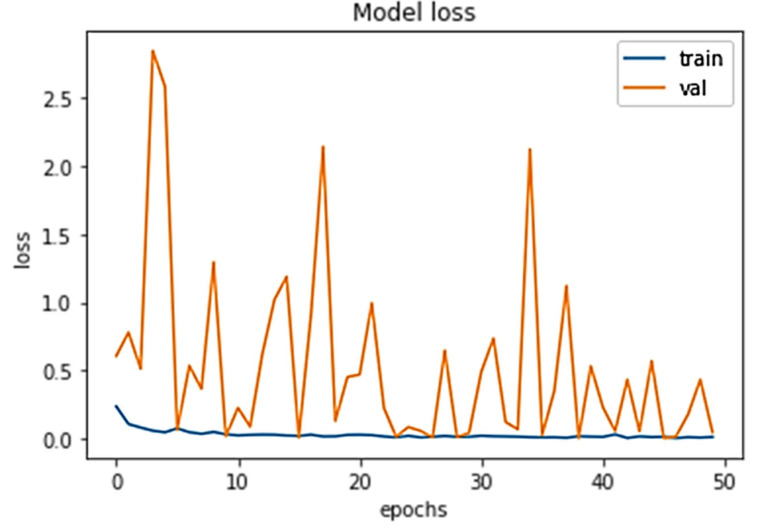
Model loss for both training and validation after 50 epochs.

**Figure 8 diagnostics-13-01703-f008:**
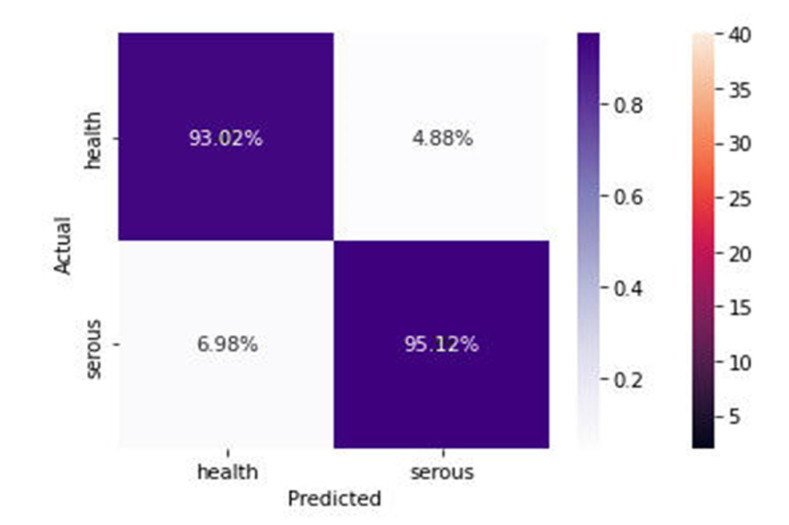
Confusion Matrix of the proposed model.

**Table 1 diagnostics-13-01703-t001:** Comparative Analysis of Machine Learning Methods for Diagnosis and Subtyping of Ovarian Cancer.

Study	Methodology	Accuracy/AUC-ROC/Sensitivity/Specificity
El-Nabawy et al. (2018) [25]	Boosting and Ensemble SVM for ovarian cancer classification using gene expression and clinical datasets	Accuracy: 80%
Wang et al. (2021) [26]	CAD method using k-NN classifier and T-POT for diagnosing borderline ovarian cancer by analyzing S-HG images	Average Accuracy: 0.976–0.96
Sawyer et al. (2021) [29]	MPM and WFI imaging techniques with Linear Discriminant Analysis classification algorithm for ovarian imaging	Accuracy: 66.66%/87.50%/62.5%
Lu et al. (2021) [32]	Tumour Origin and Assessment Method via Deep Learning (TOAD)	AUC-ROC: 0.988
Booma et al. (2020) [33]	Max-pooling enhanced with ML algorithms	Accuracy: 89%
Wen et al. (2016) [34]	Custom collection of 3D filters	AU-ROC: 83–90%
Huttunen et al. (2018) [35]	Deep neural networks for classifying unstained tissue multiphoton microscopy images	Sensitivity: 95%/Specificity: 97%
Wang et al. (2020) [36]	Two-level deep transfer learning approach	Accuracy: 87.54%
Yu et al. (2020) [37]	Radiomics model based on a convolutional neural network	AUC: 0.955–0.975
Guo et al. (2020) [41]	Deep and machine-learning techniques for clustering and classification in ovarian cancer subtyping using gene expression features	N/A
Kasture et al. (2021) [42]	DL approach for predicting and classifying ovarian cancer subtypes using histopathological images	N/A
Wu et al. (2018) [11]	DL approach to classifying OC tumors using ultrasound images	N/A
Vazquez et al. (2018) [3]	Bayesian change point for interpretation and RNN for classification	N/A
Kavitha et al. (2021) [43]	F-R-CNN to classify ovarian images	N/A
Ghoniem et al. (2021) [44]	Hybrid DL using multimodal data, combining gene and histopathology images with ALO-optimized LSTM and CNN networks	N/A
Xiao et al. (2022) [45]	Multi-omics to identify biomarkers for early OC diagnosis	N/A

**Table 2 diagnostics-13-01703-t002:** Hyperparameters and Configuration Settings of CNN for Image Classification.

Hyperparameter	Value
Number of convolutional layers	2
Kernel size	3 × 3
Number of filters	32, 64
Pooling type	Max pooling
Pooling size	2 × 2
Number of neurons in fully connected layer	128
Activation function	ReLU
Output activation function	Softmax
Loss function	Cross-entropy
Optimizer	Adam
Learning rate	0.001
Batch size	32
Number of epochs	50

**Table 3 diagnostics-13-01703-t003:** The number of epochs and the corresponding training and validation accuracies.

Epochs	Training Accuracy	Validation Accuracy
10	99.03	94.43
20	99.46	85.16
30	99.57	90.45
40	99.82	98.78
50	99.52	99.91

**Table 4 diagnostics-13-01703-t004:** Classification Report for Health and Serous Classifications.

	Precision	Recall	F1-Score	Support
Health	0.95	0.93	0.94	5520
Serous	0.93	0.93	0.94	5520
Accuracy			0.94	11,040
Macro Avg	0.94	0.94	0.94	11,040
Weighted Avg	0.94	0.94	0.94	11,040

**Table 5 diagnostics-13-01703-t005:** Comparison of Accuracy scores for different models.

Model	Accuracy
Deep hybrid learning	0.907
Convolutional Neural Network	0.897
GoogleNet (V3)	0.925
Linear Discriminant Analysis Classification	0.666

## Data Availability

Dataset will be available upon request.

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
