# Peer review of "A Deep Learning Framework for the Prediction and Diagnosis of Ovarian Cancer in Pre- and Post-Menopausal Women"

_diagnostics, 2023, doi:10.3390/diagnostics13101703_

Round 1
Reviewer 1 Report
In this paper by Blessed Ziyambe et al., the authors try to use a deep learning algorithm in order to evaluate the possibility of predicting and diagnosing ovarian carcinoma.
The underlying idea is very interesting and topical, this approach being used in the era of artificial intelligence (AI) applied to histopathological diagnostics. I suggest, however, that the introduction be entirely rewritten, as it reads like a newspaper article rather than a scientific paper; moreover, there are many linguistic inaccuracies and typos. In addition, I suggest the authors put more emphasis on the histopathological features peculiar to ovarian carcinoma, with particular attention to these two papers that could improve the quality of this manuscript:
1)Arezzo, F.; Loizzi, V.; La Forgia, D.; Moschetta, M.; Tagliafico, A.S.; Cataldo, V.; Kawosha, A.A.; Venerito, V.; Cazzato, G.; Ingravallo, G.; Resta, L.; Cicinelli, E.; Cormio, G. Radiomics Analysis in Ovarian Cancer: A Narrative Review. Appl. Sci. 2021, 11, 7833. https://doi.org/10.3390/app11177833
Arezzo, F.; Cazzato, G.; Loizzi, V.; Ingravallo, G.; Resta, L.; Cormio, G. Peritoneal Tuberculosis Mimicking Ovarian Cancer: Gynecologic Ultrasound Evaluation with Histopathological Confirmation. Gastroenterol. Insights 2021, 12, 278-282. https://doi.org/10.3390/gastroent12020024.
Author Response
Original Manuscript ID: ID: diagnostics-2315802
Original Article Title: A Deep Learning Framework for the Prediction and Diagnosis
of Ovarian cancer in Pre- and Post-Menopausal Women
To: Editor in Chief,
MDPI, Diagnostics
Re: Response to reviewers
Dear Editor,
Many thanks for insightful comments and suggestions of the referees. Thank you for
allowing a resubmission of our manuscript, with an opportunity to address the reviewers’
comments.
We are uploading (a) our point-by-point response to the comments (below) (response to
reviewers), (b) an updated manuscript with yellow highlighting indicating changes, and (c) a
clean updated manuscript without highlights (PDF main document).
By following reviewers’ comments, we made substantial modifications in our paper to
improve its clarity and readability. In our revised paper, we represent the improved
manuscript.
We have made the following modifications as desired by the reviewers:
Best regards,
Corresponding Author,
Dr. Qaisar Abbas (On behalf of authors),
Professor

Reviewer 2 Report
This study developed convolutional neural network algorithm. The model was trained using the histopathological image dataset. The dataset was split into training and validation sets. The images were first augmented before training. The model achieved an accuracy of 94%; in the confusion matrix, 95.12% were correctly classified as having a cancer infection, and 93.02% were classified as healthy cells. Here are the specific comments on this article:
1. The reviewer noted the presence of a number of grammar, tense, and punctuation errors and the English is not of publication quality, which requires improvement at the grammatical and typographical levels. The quality of the manuscript will improve after revision by a native English speaker or a professional English editing.
2. For the abstract, the reviewer suggested that instead of address the drawbacks of human expert examiners, the authors should state the advantages of this deep learning Framework.
3. The introduction section and related work section should be shortened and summarized. And some contents could also be moved to the discussion section.
4. Please make the figures clearer.
5. Please add figure legends.
6. Please cite the figures in the corresponding sentences of the manuscript.
Author Response

(The authors gave the same response as above.)

Reviewer 3 Report
The presented manuscript has attempt to predict and diagnose ovarian cancer were made using a developed convolutional neural network algorithm using the histopathological image dataset. The manuscript has addressed a good issue, but the structure of the manuscript has many problems. There are many grammatical problems and the overall structure of the manuscript should be improved. The following comments and questions would require clarification before acceptation
*** Detailed comments and questions ***
# Abstarct
1. There are lots of grammatical errors in the abstract, “Endometrioid-Ovarian-Cancer (E-O-C), and are the five”, and “is a misleading venture because this is so because an increased”. please correct this sentence.
2. What is the innovation of this manuscript? Please specify it.
3. The literature review of the abstract is not well done and all the studies reviewed are from 2018 and earlier. Authors should also review recent works in this area.
# Related Work
4. There is no figure 3 in the manuscript. Other figures are not mentioned in the literature of the manuscript. The text font of the figure 4 is too big and the resolution of the figure 5 is too low. Please modify these.
5. In the literature review section, the studies have been reviewed in a scattered manner and cannot be considered. In addition, no conclusions have been drawn at the end of this section. In addition to rewriting this section, the authors should have a conclusion at the end of the section that explains despite these works what is new in this manuscript.
# Materials and method
6. What is the dataset used to train the network? The authors should mention the dataset in more detail in the manuscript.
7. What the authors mean by “Convolution is a 528 special line function used to extract a feature in which the same small numbers”. Please justify this.
8. The structure of the network is not clear at all. The authors have not mentioned anything about the network structure used and have only raised axioms about the network. How is the training and testing process done? What was the pre-processing of the input data to the network? what is the structure of the trained network?
# Results
9. In Table 3, The authors mentioned “Deep Hybrid Learning”, what do they mean by this? What is the structure of this network?
# discussion
10. There is no proper discussion in the manuscript. The results of the study have been presented but there has been no discussion about them, what do these results show and what advantages will they create compared to the other works?
# Conclusion
11. The conclusion is not written properly, first, the authors have to restate their hypothesis or research question and then restate the findings of the manuscript. In the following, they have to tell the reader what contribution the study has made to the existing literature. Please modify the conclusion.
Author Response

(The authors gave the same response as above.)

Round 2
Reviewer 1 Report
Paper improved and suitable for pubblication.
Reviewer 3 Report
The manysceipt in the current form is acceptable.